

**Development and evaluation of the interactive Model for Air Pollution and Land Ecosystems (iMAPLE) version 1.0**

Xu Yue[1#], Hao Zhou[1#], Chenguang Tian[1], Yimian Ma[2], Yihan Hu[1], Cheng Gong[2], Hui Zheng[3], Hong Liao[1]

[1]Jiangsu Key Laboratory of Atmospheric Environment Monitoring and Pollution Control, Collaborative Innovation Center of Atmospheric Environment and Equipment Technology, School of Environmental Science and Engineering, Nanjing University of Information Science & Technology (NUIST), Nanjing, 210044, China

[2] Department Biogeochemical Integration, Max Planck Institute for Biogeochemistry, Jena, 07745, Germany

[3] Key Laboratory of Regional Climate-Environment Research for Temperate East Asia, Institute of Atmospheric Physics, Chinese Academy of Sciences, Beijing, 100029, China

Corresponding authors: Xu Yue (yuexu@nuist.edu.cn)
                         Hong Liao (hongliao@nuist.edu.cn)

[#] These authors contribute equally





**Abstract**

Land ecosystems are important sources and sinks of atmospheric components. In turn, air pollutants affect the exchange rates of carbon and water fluxes between ecosystems and atmosphere. However, these biogeochemical processes are usually not well presented in the Earth system models, limiting the explorations of interactions between land ecosystems and air pollutants from the regional to global scales. Here, we develop and validate the **i**nteractive **M**odel for **A**ir **P**ollution and **L**and **E**cosystems (iMAPLE) by upgrading the Yale Interactive terrestrial Biosphere model with process-based water cycles, fire emissions, wetland methane ($CH_4$) emissions, and the trait-based ozone ($O_3$) damages. Within the iMAPLE, soil moisture and temperature are dynamically calculated based on the water and energy balance in soil layers. Fire emissions are dependent on dryness, lightning, population, and fuel load. Wetland $CH_4$ is produced but consumed through oxidation, ebullition, diffusion, and plant-mediated transport. The trait-based scheme unifies $O_3$ sensitivity of different plant functional types (PFTs) with the leaf mass per area. Validations show correlation coefficients ($R$) of 0.59-0.86 for gross primary productivity (GPP) and 0.57-0.84 for evapotranspiration (ET) across the six PFTs at 201 flux tower sites, and yield an average $R$ of 0.68 for $CH_4$ emissions at 44 sites. Simulated soil moisture and temperature match reanalysis data with the high $R$ above 0.86 and low normalized mean biases (NMB) within 7%, leading to reasonable simulations of global GPP ($R$=0.92, NMB=1.3%) and ET ($R$=0.93, NMB=-10.4%) against satellite-based observations for 2001-2013. The model predicts an annual global area burned of 507.1 Mha, close to the observations of 475.4 Mha with a spatial $R$ of 0.66 for 1997-2016. The wetland $CH_4$ emissions are estimated to be 153.45 Tg [$CH_4$] $yr^{-1}$ during 2000-2014, close to the multi-model mean of 148 Tg [$CH_4$] $yr^{-1}$. The model also shows reasonable responses of GPP and ET to the changes in diffuse radiation, and yields a mean $O_3$ damage of 2.9% to global GPP. The iMAPLE provides an advanced tool for studying the interactions between land ecosystem and air pollutants.

**Keywords:** carbon fluxes, water cycle, fire emissions, methane emissions, ozone damage, diffuse radiation.



## 1. Introduction

As an important component on the Earth, land ecosystems regulate global carbon and water cycles. Every year, the ecosystem assimilates ~120 Pg (1 Pg = $10^{15}$ g) carbon from atmosphere through vegetation photosynthesis (Beer et al., 2010). However, most of these carbon uptake returns to atmosphere due to plant and soil respirations (Sitch et al., 2015), as well as other perturbations such as biomass burning and biogenic emissions (Carslaw et al., 2010; van der Werf et al., 2010), leading to a net carbon sink of only ~2 Pg C yr$^{-1}$ (Friedlingstein et al., 2022). Meanwhile, land ecosystems affect atmospheric moisture and soil wetness through both physical (e.g., evaporation and runoff) and physiological (e.g., leaf transpiration and root hydrological uptake) processes. Observations show that transpiration accounts for 80%-90% of the terrestrial evapotranspiration (ET) (Jasechko et al., 2013) and makes significant contributions to land precipitation especially over the tropical forests (Spracklen et al., 2012).

Different approaches have been applied to depict the spatiotemporal variations of ecosystem processes. The eddy covariance technique provides direct measurements of land carbon and water fluxes (Jung et al., 2011). However, the limited number and uneven distribution of ground sites results in large uncertainties in the upscaling of site-level fluxes to the global scale (Jung et al., 2020b). Satellite retrieval provides a unique tool for the continuous representations of land fluxes in both space and time (Worden et al., 2021). However, most of the ecosystem variables (e.g., gross primary productivity, GPP) can only be derived using available signals from remote sensing through empirical relationships (Madani et al., 2017). As a comparison, process-based models build physical parameterizations based on field and/or laboratory experiments and validate against the available *in situ* and satellite-based observations (Niu et al., 2011; Castillo et al., 2012). These models can be further applied at different spatial (from site to global) and temporal (from days to centuries) scales to identify the main drivers of the changes in carbon and water fluxes (Sitch et al., 2015). For example, a total of 17 vegetation models were validated and combined to predict the land carbon fluxes in the



past century (Friedlingstein et al., 2022); the ensemble mean of these models revealed
a steadily increasing land carbon sink from 1960 with the dominant contribution by
$CO_2$ fertilization.

While many studies quantified the ecosystem responses to the effects of $CO_2$, climate,
and human activities (Piao et al., 2009; Sitch et al., 2015), few have explored the
interactions between air pollution and land ecosystems. Such biogeochemical processes
become increasingly important in the Anthropocene period with significant changes in
atmospheric compositions. For example, observations found that nitrogen and
phosphorus constrain the $CO_2$ fertilization efficiency of global vegetation (Terrer et al.,
2019), but such limiting effect is ignored or underestimated in most of the current
models (Wang et al., 2020). Tropospheric ozone ($O_3$) damages plant photosynthesis and
stomatal conductance, inhibiting carbon assimilation and the ET from the land surface
(Sitch et al., 2007; Lombardozzi et al., 2015). Atmospheric aerosols can enhance
photosynthesis through diffuse fertilization effects (Mercado et al., 2009) but
meanwhile decrease photosynthesis by reducing precipitation (Yue et al., 2017). In turn,
ecosystems act as both the sources and sinks of atmospheric components. Biomass
burning emits a large amount of carbon dioxide, trace gases, and particulate matters,
further influencing air quality (Chen et al., 2021), ecosystem functions (Yue and Unger,
2018), and global climate (Tian et al., 2022). Biogenic volatile organic compounds
(BVOCs) are important precursors for both surface $O_3$ and secondary organic aerosols
(Wu et al., 2020), which can feed back to affect biogenic emissions (Yuan et al., 2016)
and carbon assimilations (Rap et al., 2018). Wetland methane ($CH_4$) emissions account
for the dominant fraction of natural sources of $CH_4$, and are projected to increase under
the global warming scenarios (Zhang et al., 2017; Rosentreter et al., 2021). On the other
hand, stomatal uptake dominates the dry deposition of air pollutants over the vegetated
land (Lin et al., 2020). Meanwhile, ET from forest results in the increase of water vapor
in atmosphere (Spracklen et al., 2012), affecting the consequent rainfall and wet
deposition of particles.






Currently, numerical models are in general developed separately for atmospheric chemistry and ecosystem processes. The chemical transport models are usually driven with prescribed emissions of biomass burning (Warneke et al., 2023) and wetland methane (Heimann et al., 2020), while the ecosystem models often ignore the biogeochemical impacts of $O_3$ and aerosols (Friedlingstein et al., 2022). In an earlier study, we developed and validated the Yale Interactive terrestrial Biosphere (YIBs) model version 1.0 with the special focus on the interactions between atmospheric chemistry and land ecosystems (Yue and Unger, 2015). Thereafter, the YIBs model has been used offline to assess the $O_3$ vegetation damage (Yue et al., 2016), aerosol diffuse fertilization (Yue and Unger, 2017), BVOCs emissions (Cao et al., 2021a), as well as coupled to other models to investigate the carbon-chemistry-climate interactions (Lei et al., 2020; Gong et al., 2021). The YIBs model has joined the multi-model intercomparison project of TRENDY since the year 2020 and showed reasonable performance in the simulation of carbon fluxes (Friedlingstein et al., 2020). However, the YIBs model failed to predict the typical hydrological variables such as ET and runoff due to the missing of carbon-water coupling modules. Furthermore, the model did not consider the nutrient limitation on plant photosynthesis and ignored some key exchange fluxes between land and atmosphere.

131

In this study, we develop the **i**nteractive **M**odel for **A**ir **P**ollution and **L**and **E**cosystems (iMAPLE) by coupling the process-based water cycle module from Noah-MP (Niu et al., 2011) to the carbon cycle in the YIBs (Figure 1). In addition, we update the original YIBs model with some major advances in the biogeochemical processes including dynamic fire emissions, wetland $CH_4$ emissions, nutrient limitations on photosynthesis, and the trait-based $O_3$ vegetation damage. The detailed descriptions of these updates are presented in the next section. The iMAPLE is fully validated against available measurements in Section 3. The last section will summarize the model performance and rethink the prospective directions for future development.






**2. Models and data**

**2.1 Main features of YIBs model**

The YIBs model is a process-based vegetation model predicting land carbon fluxes with
dynamic changes in tree height, leaf area index, and carbon pools (Yue and Unger, 2015,
thereafter YU2015). A total of nine plant functional types (PFTs) are considered
including evergreen broadleaf forest (EBF), evergreen needleleaf forest (ENF),
deciduous broadleaf forest (DBF), tundra, shrubland, $C_3/C_4$ grassland, and $C_3/C_4$
cropland. Leaf photosynthesis is calculated using the well-established Michaelis-
Menten enzyme-kinetics scheme (Farquhar et al., 1980) and is coupled to stomatal
conductance with the modulations of air humidity and $CO_2$ concentrations (Ball et al.,
1987). The model applies a two-leaf approach to distinguish the irradiating states for
sunlit and shading leaves and adopts an adaptive stratification for the radiative transfer
processes within canopy layers (Spitters, 1986). The gross carbon assimilation is further
regulated by the optimized plant phenology, which is mainly dependent on temperature
and light for deciduous trees (Yue et al., 2015) but temperature and/or moisture for
shrubland and grassland (YU2015). The assimilated carbon is allocated among leaf,
stem, and root to support autotrophic respiration and development, the latter of which
is used to update plant height and leaf area (Cox, 2001). The input of litterfall triggers
the carbon transition among 12 soil carbon pools and determines the magnitude of
heterotrophic respiration with the joint effects of soil temperature, moisture, and texture
(Schaefer et al., 2008). The net carbon uptake is then calculated by subtracting
ecosystem respiration (plant and soil) and environmental perturbations (reforestation or
deforestation) from the gross carbon assimilation (Yue et al., 2021). The YIBs model
reasonably reproduces the observed spatiotemporal patterns of global carbon fluxes and
makes contributions to the Global Carbon Project with the long-term simulations of
land carbon sink in the past century (Friedlingstein et al., 2020). The model specifically
considers air pollution impacts on land ecosystems (Figure 1), such as the ozone
vegetation damage (Yue and Unger, 2014) and aerosol diffuse fertilization effect (Yue



and Unger, 2017). The YIBs implements two different schemes for BVOCs emissions
(Figure 1), including the Model of Emissions of Gases and Aerosols from Nature
(MEGAN, Guenther et al., 2012) and the photosynthesis-dependent (PS_BVOC)
scheme (Unger et al., 2013).

**2.2 New processes in iMAPLE model**
2.2.1 Process-based water cycles
We implement the hydrological module from Noah-MP into the iMAPLE model (Niu
et al., 2011). The water budget closure is achieved by constructing water-balance
equations among precipitation ($P$, Kg m$^{-2}$ s$^{-1}$), evapotranspiration ($ET$, Kg m$^{-2}$ s$^{-1}$),
runoff, and terrestrial water storage change ($\Delta TWS$) on each grid cell as follows:
$$P = ET + runoff + \Delta TWS \qquad (1)$$

Here, hourly $P$ from MERRA-2 reanalyses is used as the input.

We then divide $ET$ into three portions including plant transpiration ($TRA$), canopy
evaporation ($ECAN$) and ground evaporation ($EGRO$):
$$ET = TRA + ECAN + EGRO \qquad (2)$$

For vegetated grids, $TRA$ is calculated as follows:
$$TRA = \frac{\rho_{air} \cdot CP_{air} \cdot C_{tra} \cdot (e_{sat} - e_{ca})}{PC} \qquad (3)$$

where $\rho_{air}$ is air density, $CP_{air}$ is heat capacity of dry air, and $PC$ is the
psychrometric constant. $e_{sat}$ is the saturated vapor pressure at the leaf temperature,
$e_{ca}$ is the vapor pressure of the canopy air and $C_{tra}$ is leaf transpiration conductance,
which is calculated based on the Ball-Berry scheme of stomatal resistance (Yue and
Unger, 2015).

Runoff includes surface ($R_{srf}$) and subsurface ($R_{sub}$) components:
$$runoff = R_{srf} + R_{sub} \qquad (4)$$

The surface runoff is calculated as follows:
$$R_{srf} = Q_{soil,srf} - Q_{soil,in} \qquad (5)$$





where $Q_{soil,srf}$ is the incident water in the soil surface and is the sum of the
precipitation, snowmelt and dewfall. Here, we assume independent and exponential
distributions of infiltration capacity and precipitation in each grid cell when considering
soil infiltration processes and $Q_{soil,in}$ is the infiltration into the soil, following the
approach by Schaake et al. (1996). We assume free drainage processes in the soil
column bottom, thus the $R_{sub}$ is calculated as follows:
$$R_{sub} = \alpha_{slope} \cdot K_4 \tag{6}$$

where $\alpha_{slope} = 0.1$ is the terrain slope index. $K_4$ is the hydraulic conductivity in the
bottom soil layer from soil parameterizes used in Clapp and Hornberger (1978).

Terrestrial water storage ($TWS$) is the sum of groundwater storage ($W_{gw}$), soil water
content ($W_{soil}$) and snow water equivalent ($W_{snow}$):
$$TWS = W_{gw} + W_{snow} + \sum_{i=1}^{N_{soil}} W_{soil} \tag{7}$$

Here, the soil module includes four layers ($N_{soil}= 4$) and $W_s$ is calculated by the
volumetric water content ($W_i$) as follows:
$$W_s = \rho_{wat} \cdot W_i \cdot \Delta Z_i \quad for \ i = 1, 2, 3, 4 \tag{8}$$

where water density ($\rho_{wat}$) $= 1000 \ kg \ m^{-3}$, and $\Delta Z_i$ $= 0.1, 0.3, 0.6$ and $1m$, respectively.
Hourly $W_i$ depends on variations of soil water diffusion ($D$) and hydraulic
conductivity ($K$) as follows:
$$\frac{\partial W}{\partial t} = \frac{\partial}{\partial z}\left(D \frac{\partial W}{\partial z}\right) + \frac{\partial K}{\partial z} \tag{9}$$

Here, $K$ and $D$ are calculated following the parameterizations of Clapp-Hornberger
curves (Clapp and Hornberger, 1978):
$$\frac{K}{K_{sat}} = \left(\frac{W}{W_{sat}}\right)^{2b+3} \tag{10}$$

$$D = K \cdot \frac{\partial \varphi}{\partial W} \tag{11}$$

$$\frac{\varphi}{\varphi_{sat}} = \left(\frac{W}{W_{sat}}\right)^{-b} \tag{12}$$

where $\varphi_{sat}$, $W_{sat}$ and $K_{sat}$ are saturated soil capillary potential, volumetric
water content and hydraulic conductivity. Exponent $b$ is an empirical constant





depending on soil types. Soil moisture is calculated as the ratio of $W_s$ to $W_{sat}$.

Soil temperature ($T_s$) is calculated through physical processes as follows:
$$\frac{\partial T_s}{\partial t} = \frac{1}{C}\frac{\partial}{\partial z}\left(K_T \frac{\partial T_s}{\partial z}\right) \tag{13}$$

Here $K_T$ is soil specific heat capacity:
$$K_T = K_e \cdot \left(K_s - K_{dry}\right) + K_{dry} \tag{14}$$

where $K_e$, $K_s$ and $K_{dry}$ are Kersten values as a function of soil wetness, saturated
soil heat conductivity and that under dry air conditions (Niu et al., 2011). $C$ in Equation
(13) is the specific heat
$$C = W_{lip} \cdot C_{lip} + W_{ice} \cdot C_{ice} + (1 - W_{sat}) \cdot C_{sat} + (W_{sat} - W) \cdot C_{air} \tag{15}$$

Here, $W_{lip}$, $C_{lip}$ and $W_{ice}$, $C_{ice}$ indicate water content and heat capacity on soil
water and ice. $C_{sat}$ and $C_{air}$ are saturated and air heat capacity, which are empirical
constants (Niu et al., 2011).

2.2.2 Dynamic fire emissions
We implement the active global fire parameterizations from Pechony and Shindell
(2009) and Li et al. (2012) to the iMAPLE model. The fire emissions are determined
by several key factors such as fuel flammability, natural ignitions, human activities, and
fire spread. The fire count N_fire depends on flammability (*Flam*), fire ignition (including
both natural ignition rate $I_N$ and anthropogenic ignition rate $I_A$) and anthropogenic
suppression (*F_NS*):
$$N_{fire} = Flam \times (I_N + I_A) \times F_{NS} \tag{16}$$

*Flam* is a unitless metric representing conditions conducive to fire occurrence. It is
parameterized as a function of vapor pressure deficit (VPD), precipitation (Prec), and
leaf area index (LAI):
$$Flam = VPD \times e^{-2 \times Prec} \times LAI \tag{17}$$

$I_N$ depends on the cloud-to-ground lightning and $I_A$ can be expressed as:
$$I_A = 0.03 \times PD \times k(PD) \tag{18}$$

where *PD* is population density. The empirical function of $k(PD) = 6.8 \times PD^{-0.6}$ stands





for ignition potentials by human activity. The fraction of non-suppressed fires $F_{NS}$ is
derived as:

$$F_{NS} = 0.05 + 0.95 \times e^{-0.05 \times PD} \tag{19}$$


The burned area of a single fire ($BA_{single}$) is typically taken to be elliptical in shape
associated with near-surface wind speed ($U$) and relative humidity ($RH$):

$$BA_{single} = \frac{\pi \times UP^2}{4 \times LB} \times (1 + \frac{1}{HB})^2 \tag{20}$$

where $LB$ and $HB$ are length-to-breadth ratio and head-to-back ratio, respectively:

$$LB = 1 + 10 \times (1 - e^{-0.06 \times U}) \tag{21}$$

$$HB = \frac{LB + (LB^2 - 1)^{0.5}}{LB - (LB^2 - 1)^{0.5}} \tag{22}$$

The rate of fire spread ($UP$) is computed as:

$$UP = UP_{max} \times f_{RH} \times f_{\theta} \times G(W) \tag{23}$$

Here, $UP_{max}$ is the maximum fire spread rate depending on PFTs, $f_\theta$ is set to 0.5 and
$f_{RH}$ is calculated as:

$$f_{RH} = \begin{cases} 0, & RH \leq RH_{low} \\ \frac{RH_{up} - RH}{RH_{up} - RH_{low}}, & RH_{low} < RH < RH_{up} \\ 1, & RH \geq RH_{up} \end{cases} \tag{24}$$

In this study, we set $RH_{low}$ =30 % and $RH_{up}$ =70 %. G(W) is the limit of the fire spread:

$$G(W) = \frac{LB}{1 + \frac{1}{HB}} \tag{25}$$


Finally, the burned aera ($BA$) is represented as:

$$BA = BA_{single} \times N_{fire} \tag{26}$$

The fire-emitted trace gases and aerosols ($Emis$) are calculated as:

$$Emis = BA \times EF \tag{27}$$

where $EF$ is the emission factors for different species (such as black carbon and organic
carbon aerosols).

2.2.3 Wetland methane emissions
We implement the process-based wetland $CH_4$ emissions into the iMAPLE model. For
each soil layer, the flux of CH4 ($F_{CH_4}$) is calculated as the difference between production
($P_{CH_4}$) and consumptions, which include oxidation ($O_{CH_4}$), ebullition ($E_{CH_4}$), diffusion
($D_{CH_4}$), and plant-mediated transport through aerenchyma ($A_{CH_4}$) as follows:
$$F_{CH_4} = P_{CH_4} - O_{CH_4} - E_{CH_4} - D_{CH_4} - A_{CH_4} \tag{28}$$

The net methane emission to the atmosphere is the sum of ebullition, diffusion and
aerenchyma transport from the top soil layer.

The production of CH4 in soil depends on the quantity of carbon substrate and
environmental conditions including soil temperature $T_s$, pH, and wetland inundation
fraction $f_{wetland}$ as follows:
$$P_{CH_4} = R_h r f_{Ts} f_{pH} f_{wetland} \tag{29}$$

where $R_h$ is the heterotrophic respiration estimated at the grid cell ($mol\ C\ m^{-2}\ s^{-1}$).
$r$ represents the release ratio of methane and carbon dioxide (Wania et al., 2010). We
determine the dependence on $T_s$ and soil pH in iMAPLE based on the parameterizations
from the TRIPLEX-GHG model (Zhu et al., 2014). For the temperature-dependence,
the $Q_{10}$ relationships are applied as follows:
$$Q_{10} = r_b Q_b^{\frac{T_s - T_{base}}{10}} \tag{30}$$

Here $r_b$ is set to 3.0 and $Q_b$ is 1.33 with a base temperature ($T_{base}$) of 25°C (Zhu et al.,
2014; Paudel et al., 2016). The inundation fraction of wetland at each cell describes the
proportion of anaerobic conditions (Zhang et al., 2021). We ignore the impact of redox
potential (Eh) because global observations are not available and the Eh-related
processes are poorly characterized in current models (Wania et al., 2010).

The oxidation of CH4 is a series of aerobic activities related to temperature and CH4
concentrations:
$$O_{CH_4} = [CH_4] f_{Ts} f_{CH_4} \tag{31}$$

where $[CH_4]$ is the methane amount in each soil layer ($gCm^{-2}layer^{-1}$). $f_{CH4}$ is the
CH4 concentration factor representing a Michaelis-Menten kinetic relationship:
$$f_{CH4} = \frac{[CH_4]}{[CH_4] + K_{CH}} \tag{32}$$





where $K_{CH4} = 5\ \mu mol\ L^{-1}$ is the half-saturation coefficient with respect to CH$_4$ (Walter
and Heimann, 2000). For temperature-dependence of oxidation, the $Q_{10}$ relationship
with $r_b = 2.0$, $Q_b = 1.9$, and $T_{base} = 12°C$ is adopted (Zhu et al., 2014; Paudel et al., 2016).

The diffusion of CH$_4$ follows the Fick's law with dependence on CH$_4$ concentrations
and the molecular diffusion coefficients of CH$_4$ in the air ($D_a = 0.2\ cm^2 s^{-1}$) and water
($D_w = 0.00002\ cm^2 s^{-1}$) respectively (Walter and Heimann, 2000). For each soil layer
$i$, the diffusion coefficient $D_i$ can be calculated as follows :
$D_i = D_a \times \left(R_{sand} \times 0.45 + R_{silt} \times 0.2 + R_{clay} \times 0.14\right) \times f_{tort} \times S_{poro} \times (1 -$
$WFPS_i) + D_w \times WFPS_i$           (33)
where $R_{sand}$, $R_{silt}$, $R_{clay}$ is the relative content of sand, silt, and clay in the soil,
$f_{tort} = 0.66$ is tortuosity coefficient, $S_{poro}$ is soil porosity, and $WFPS$ represents the
pore space full of water (Zhuang et al., 2004).

The ebullition of CH$_4$ occurs when CH$_4$ concentration is above the threshold of 0.5
$mol\ CH_4 m^{-3}$ (Walter et al., 2001). Since the process of ebullition occurs in a very short
time, the bubbles will generate at once and all the flux will be released to atmosphere
if the concentration reaches the threshold. The plant-mediated transport of CH$_4$ through
aerenchyma is dependent on the concentration gradient of CH$_4$ and the plant-related
factors (Zhu et al., 2014).

2.2.4 The down regulation on photosynthesis
We implement the down regulation parameterization from Arora et al. (2009) to indicate
the nutrient limitations on leaf photosynthesis. A down-regulating factor $\varepsilon$ is calculated
as a function of CO$_2$ concentrations ($C$) as follows:
$$\varepsilon(C) = \frac{1 + \gamma_{gd}\ln{(C/C_0)}}{1 + \gamma_g \ln{(C/C_0)}}$$       (34)
where $C_0$ is a reference CO$_2$ concentration set to 288 ppm. The values of $\gamma_{gd} = 0.42$ and
$\gamma_g = 0.90$ are derived from multiple measurements to constrain the CO$_2$ fertilization.
Then the down-regulated photosynthesis is calculated by scaling the original value with





the factor of $\varepsilon$.

2.2.5 Trait-based $O_3$ vegetation damaging scheme
The YIBs model considers $O_3$ vegetation damage using the flux-based scheme proposed
by Sitch et al. (2007) (thereafter S2007), which determines the damaging ratio $F$ of
plant photosynthesis as follows:
$$F = a_{PFT} \times max\{f_{O3} - t_{PFT}, 0\} \quad (35)$$

Here, the $f_{O3}$ denotes $O_3$ stomatal flux (nmol m$^{-2}$ s$^{-1}$) defined as:
$$f_{O3} = \frac{[O_3]}{r + \left[\frac{k_{O3}}{g_p \times (1-F)}\right]} \quad (36)$$

where $[O_3]$ represents the $O_3$ concentrations at the reference level (nmol m$^{-3}$). $r$ is the
sum of boundary and aerodynamic resistance between leaf surface and reference level
(s m$^{-1}$). $g_p$ is the potential stomatal conductance for $H_2O$ (m s$^{-1}$). $k_{O3} = 1.67$ is a
conversion factor of leaf resistance for $O_3$ to that for water vapor. The level of $O_3$
damage is then determined by the PFT-specific sensitivity $a_{PFT}$ and threshold $t_{PFT}$,
which are different among PFTs.

In iMAPLE, we implement the trait-based $O_3$ vegetation damaging scheme to unify the
inter-PFT sensitivities (Ma et al., 2023):
$$a_{PFT} = \frac{a}{LMA} \quad (37)$$

Here, a unified plant sensitivity $a$ (nmol$^{-1}$ g s) is scaled by leaf mass per area (LMA, g
m$^{-2}$) to derive the sensitivity of a specific PFT ($a_{PFT}$). Accordingly, the damaging
fraction $F$ is modified as follows:
$$F = a \times max\left\{\frac{f_{O3}}{LMA} - t, 0\right\} \quad (38)$$

Here $t$ (nmol g$^{-1}$ s$^{-1}$) is a unified flux threshold for $O_3$ vegetation damage. The updated
scheme considers the dilution effects of $O_3$ dose through leaf cross-section by
incorporating LMA. Plants with high LMA (e.g., ENF and EBF) usually have low
sensitivities, and those with low LMA (e.g., DBF and crops) are more sensitive to $O_3$
damages. The unified sensitivity $a$ is set to 3.5 nmol$^{-1}$ g s and threshold $t$ is set to 0.019



nmol g$^{-1}$ s$^{-1}$ by calibrating simulated $F$ values with literature-based measurements (Ma
et al., 2023).

**2.3 Design of simulations**
We perform four sensitivity experiments with the iMAPLE model. The baseline (BASE)
simulation considers the two-way coupling between carbon and water cycles, so that
the prognostic soil meteorology drives canopy photosynthesis and evapotranspiration.
A sensitivity run named BASE_NW is set up by turning off the water cycle in the
iMAPLE model. In this simulation, the soil moisture and soil temperature are adopted
from the Modern-Era Retrospective Analysis for Research and Applications, Version 2
(MERRA-2) reanalyses (Gelaro et al., 2017). The third and fourth runs turn on the $O_3$
vegetation damage effect using either the LMA-based scheme (O3LMA) or the S2007
scheme (O3S2007). For all simulations, the iMAPLE model is driven with the hourly
surface meteorology at a spatial resolution of 1º×1º from the MERRA-2 reanalyses,
including surface air temperature, air pressure, specific humidity, wind speed,
precipitation, snowfall, shortwave and longwave radiation. We run the model for the
period of 1980-2021 using the initial conditions of the equilibrium soil carbon pool,
tree height, and water fluxes from a spin-up run of 200 years.

The iMAPLE model is driven with observed $CO_2$ concentrations from Mauna Loa
(Keeling et al., 1976) and the land cover fraction of nine PFTs derived by combining
satellite retrievals from both Moderate Resolution Imaging Spectroradiometer (MODIS)
(Hansen et al., 2003) and Advanced Very High Resolution Radiometer (AVHRR)
(Defries et al., 2000). For fire emissions, we use Gridded Population of the World
version 4 (https://sedac.ciesin.columbia.edu/data/collection/gpw-v4) to calculate
human ignition and suppression. The lighting ignition is calculated using the flash rate
from Very High Resolution Gridded Lightning Climatology Data Collection Version 1
(https://ghrc.nsstc.nasa.gov/uso/ds_details/collections/lisvhrcC.html).  For  wetland
$CH_4$ emissions, we use the 2000-2020 global dataset of Wetland Area and Dynamics





for Methane Modeling (WAD2M) derived from static datasets and remote sensing
(Zhang et al., 2021), global soil pH from Hengl et al. (2017), and gridded soil texture
from Scholes et al. (2011). For the LMA-based $O_3$ damage scheme, we use gridded
LMA derived from the trait-level dataset of TRY (Kattge et al., 2011) using the random
forest model (Moreno-Martínez et al., 2018).

**2.4 Data for validations**
We use observational datasets to validate the biogeochemical processes and related
variables simulated by the iMAPLE model. For simulated carbon and water fluxes, site-
level observations are collected from the 201 sites at the FLUXNET network (Table
S1). We also use the global gridded observations of GPP from the satellite retrievals
including the solar-induced chlorophyll fluorescence (SIF) product GOSIF (Li and
Xiao, 2019) and the Global land surface satellite (GLASS) product (Yuan et al., 2010).
The global observations of ET are adopted from the benchmark product of FLUXCOM
(Jung et al., 2020a) and the satellite-based GLASS product. For the dynamic fire
module, we use monthly observed area burned from the Global Fire Emission Database
version 4.1 with small fires (GFED4.1s) during 1997-2016 (van der Werf et al., 2010;
Randerson et al., 2012). For methane emissions, we use site-level measurements of $CH_4$
fluxes from the FLUXNET-$CH_4$ network (Delwiche et al., 2021). We exclude the
monthly records with missing data at more than half of the days and calculate the long-
term mean fluxes for the seasonal cycle. In total, we select 44 sites with at least six
months of data available for the validations (Table S2). We also use the anthropogenic
sources of $CH_4$ from the archive of Coupled Model Intercomparison Project phase 6
(CMIP6, https://esgf-node.llnl.gov/projects/input4mips/).

**3. Model evaluations**
3.1 Site-level evaluations
We compare the simulated carbon and water fluxes to *in situ* measurements at 201
FLUXNET sites (Figure 2). Among these sites, 95 are tree species with the major PFT





of ENF and 106 are non-tree species with the maximum number for shrubland. Most
(71%) of sites are located at the middle latitudes (30º-60ºN) of the Northern Hemisphere
(NH), especially in the U.S. and Europe. Compared to the earlier evaluations in
YU2015, we have much more sites in the tropics (22 in this study vs. 5 in YU2015),
Asia (20 in this study vs. 1 in YU2015), and Southern Hemisphere (28 in this study vs.
7 in YU2015) in this study.

Simulated GPP shows correlation coefficients (R) of 0.59-0.86 for the six main PFTs
with varied sample numbers (Figure 3). The highest R is achieved for ENF, though the
model underestimates the mean GPP magnitude by 20.62% for this species. On average,
simulated GPP is lower than observations for most PFTs. Compared to the YIBs model,
iMAPLE with coupled water cycle improves the GPP simulations for ENF and
grassland but worsens the predictions for other species. The main cause of such deficit
is the application of MERRA-2 reanalyses in the iMAPLE simulations instead of the
site-level meteorology used in the YU2015. The biases in the meteorological input may
cause uncertainties in the simulation of GPP fluxes (Ma et al., 2021). Furthermore, the
increase of site number and record length may decrease the R to some extent.

Simulated ET matches observations with correlation coefficients of 0.57-0.84 at the
FLUXNET sites (Figure 4). Relatively better performance is achieved for ENF (R=0.83)
and grassland (R=0.84), for which the model yields good predictions of GPP as well.
In contrast, low correlations and high biases are predicted for shrubland and cropland.
For the shrubland sites, different land types (e.g., closed shrublands, permanent
wetlands, and woody savannas) share the same parameters in the iMAPLE model,
resulting in the biases in depicting the site-specific carbon and water fluxes. For
cropland, the prognostic phenology of grass species is applied in the model due to the
missing of plantation information for individual sites. Even with these deficits, the
iMAPLE model in general captures the spatiotemporal variations of GPP and ET at
most sites.






We further compare the simulated wetland $CH_4$ fluxes with observations at the
FLUXNET-$CH_4$ sites. Similar to the carbon flux sites, most of these $CH_4$ flux sites are
located in the NH (Figure 5a). However, different from the carbon fluxes which usually
range from 0 to 15 g C m$^{-2}$ day$^{-1}$, the $CH_4$ fluxes show a wide range across several
orders of magnitude from $10^{-2}$ to $10^{3}$ g [$CH_4$] m$^{-2}$ yr$^{-1}$ (Figure 5b). Such a large contrast
requires a more realistic configuration of model parameters to distinguish the large
gradient among sites. For example, US-Tw1 and US-Twt are two nearby sites within a
distance of 1 km. However, average $CH_4$ flux shows a difference of 3.7 times with 66.31
g[$CH_4$] m$^{-2}$ yr$^{-1}$ in US-Tw1 and 18.16 g[$CH_4$] m$^{-2}$ yr$^{-1}$ in US-Tw4 during 2011-2017. In
the model, these two sites share the same land surface properties because they are
located on the same grid. On average, simulated $CH_4$ fluxes are correlated with
observations at a moderate R of 0.68 and a normalized mean bias (NMB) of -28%.

3.2 Grid-level evaluations
The coupling of Noah-MP module enables the dynamic prediction of soil parameters
by the iMAPLE model. We compare the simulated soil moisture and soil temperature
with MERRA-2 reanalyses (Figure 6). Both simulations (Figure 6a) and observations
(Figure 6b) show low soil moisture over arid and semi-arid regions with the minimum
in North Africa. The model also captures the high soil moisture in tropical rainforest.
However, the prediction underestimates soil moisture in boreal regions in NH (Figure
6c). On the global scale, simulated soil moisture matches observations with a high R of
0.86 and a low NMB of -6.9%. These statistical metrics are further improved for the
simulated soil temperature with the R of 0.99 and NMB of 0.5% against observations
(Figure 6f). The simulation reproduces the observed spatial pattern with a uniform
warming bias.

Driven with the prognostic soil moisture and temperature, the iMAPLE model predicts
reasonable land carbon and water fluxes (Figure 7). Simulated GPP (Figure 7a)





reproduces observed patterns (Figure 7b) with high values in the tropical rainforest,
moderate values in the boreal forests, and low values in the arid regions. The predicted
GPP is higher than observations over the tropical rainforest (Figure 7c). However, such
overestimation may instead be an indicator of biases in the ensemble observations,
which are derived from the empirical models instead of direct measurements (Running
et al., 2004; Yuan et al., 2010). Our site-level evaluations show that iMAPLE predicts
reasonable GPP values at the EBF sites (Figure 3). Despite this inconsistency, the model
yields a high R of 0.92 and a small NMB of 1.3% for GPP against observations on the
global scale (Figure 7c). Simulated ET (Figure 7d) matches the observations (Figure 7e)
with high values in the tropical rainforest and secondary high values in the boreal forest.
In general, the prediction is lower than observations except for the eastern U.S. and
eastern China (Figure 7f). On average, the iMAPLE model shows the R of 0.93 and
NMB of -10.4% in the simulation of ET compared to the ensemble of observations.

We further compare the simulated GPP with or without dynamic water cycle (Figure 8).
Relative to the simulations driven with MERRA-2 soil moisture and temperature, the
iMAPLE model coupled with Noah-MP water module predicts very similar GPP over
the hotspot regions such as tropical rainforest and boreal forest (Figure 8a). However,
the coupled model predicts lower GPP for grassland in the tropics (e.g., South America
and central Africa) but higher GPP in arid regions (e.g., South Africa and Australia).
Since the baseline GPP is very low in arid regions, the relative changes are even larger
than 100% over those areas. These GPP differences are mainly driven by the changes
in soil moisture, which increases over the arid regions with the dynamic water cycle
(Figure 6c). The reduction of soil moisture in the high latitudes of NH shows limited
impacts on the predicted GPP, likely because the boreal ecosystem is more dependent
on temperature than moisture (Beer et al., 2010).

3.3 Ecosystem perturbations to air pollution
Within the iMAPLE framework, the land ecosystem perturbs atmospheric components





through the emissions from biomass burning, wetland $CH_4$, and BVOCs. We compare
the simulated burned fraction with observations from GFED4.1s (Figure 9). The largest
burned fraction is predicted over the Sahel region and countries of Angola and Zambia,
surrounding the low center of Congo rainforest. Moderate burnings could be found in
northern Australia and eastern South America. Most of these hotspots are located on the
grassland and shrubland in the tropics, where the high temperature and limited rainfall
promotes regional fire activities. The model reasonably captures the observed fire
pattern with a spatial correlation of 0.66 and NMB of 6.05% (Figure 9c), though the
model overestimates the area burned in South Africa. The predicted fire area is used to
derive biomass burning emissions of air pollutants (e.g., carbon monoxide, nitrogen
oxides, black carbon, organic carbon, sulfur dioxide) with the specific emission factors
(Tian et al., 2023).

The wetland emissions of $CH_4$ show hotspots over tropical rainforests (Figure 10a),
where the dense soil carbon provides abundant substrates for emissions and the warm
climate promotes the emission rates. The secondary hotspots are located at the boreal
regions in the NH. This spatial pattern is very similar to the map of wetland $CH_4$
emissions predicted by an ensemble of 13 biogeochemical models (Saunois et al., 2020).
On the global scale, the total wetland emission is 153.45 Tg [$CH_4$] $yr^{-1}$ during 2000-
2014, close to the average of 148±25 Tg [$CH_4$] $yr^{-1}$ for 2000-2017 estimated by the
multiple models. As a comparison, anthropogenic source of $CH_4$ show the high amount
in China and India due to the large emissions from fossil fuels and agriculture (Figure
10b). On the global scale, the wetland emissions are equivalent to 45.3% of the total
anthropogenic emissions.

Isoprene emissions from the two schemes in the iMAPLE model show similar spatial
distributions with the hotspots over tropical rainforest (Figure 11), where the warm
climate and abundant light are favorable for the biogenic emissions. Compared to the
MEGAN scheme, the PS_BVOC scheme yields higher emissions in the tropical





rainforest and boreal forest, but lower emissions for the shrubland and grassland in
semiarid regions (Figure 11c). Such differences are attributed to the varied processes as
well we the emission factors. Our earlier study showed that PS_BVOC scheme predicts
stronger trends in isoprene emissions than MEGAN (Cao et al., 2021a), because the
former considers both $CO_2$ fertilization and inhibition effects while the latter considers
only the inhibition effects. On the global scale, isoprene emissions are 550 Tg yr$^{-1}$ with
PS_BVOC (Figure 11a) and 611 Tg yr$^{-1}$ with MEGAN (Figure 11b). These amounts are
higher than the ensemble mean of 448 Tg yr$^{-1}$ from the CMIP6 models (Cao et al.,
2021b), but in general within the range of 412-601 Tg yr$^{-1}$ as summarized by Carslaw
et al. (2010).

3.4. Air pollution impacts on ecosystem fluxes
We assess the damaging effects of surface $O_3$ to GPP with two schemes (Figure 12).
Simulated GPP losses show similar patterns with high damages in eastern U.S., western
Europe, and eastern China, where surface $O_3$ level is high due to the anthropogenic
emissions. Limited GPP damages are predicted in the tropics though with abundant
forest coverage due to the low level of $O_3$ pollution. Compared to the S2007 scheme,
predicted GPP loss is further alleviated in tropical rainforest with the LMA-based
scheme, because the latter scheme determines lower $O_3$ sensitivity for evergreen trees
due to their higher content of chemical resistance with the larger LMA value (Ma et al.,
2023). On the global scale, the average GPP loss is -2.9% with the LMA scheme and -
3.2% with the S2007 scheme. Such damage to GPP is weaker than the estimate of -4.8%
in Ma et al. (2023) because of the differences in $O_3$ concentrations, vegetation types,
and photosynthetic parameters.

Atmospheric aerosols cause perturbations to both direct and diffuse radiation, which
have different efficiencies in enhancing plant photosynthesis. Here, we separate the
diffuse (diffuse fraction > 0.75) and direct (diffuse fraction < 0.25) components of solar
radiation, and aggregate the GPP and ET fluxes for different radiation periods at certain





intervals (Figure 13). At the six selected sites, observed GPP is higher and grows faster
with more diffusive light than that under the direct light conditions (Figure 13a-13f).
Simulations in general reproduce such feature with the comparable variability. In the
earlier study, simulated diffuse fertilization efficiency for GPP (changes of GPP per unit
diffuse radiation) was well validated against observations at more than 20 sites (Yue
and Unger, 2018). Such amelioration of GPP suggests that moderate aerosol loading is
beneficial for ecosystem carbon uptake (Yue and Unger, 2017). However, the dense
aerosol loading may instead weaken plant photosynthesis due to the large reduction in
direct radiation .

We further evaluate the ET responses to diffuse and direct radiation from the iMAPLE
model (Figure 13g-13l). Although ET is slightly higher at the diffusive condition, the
growth rates are weaker than that of GPP. The main cause of such difference is related
to the varied light dependence of ET components, which consist of canopy evaporation
and transpiration. Transpiration is tightly coupled with photosynthesis and will increase
by diffuse radiation at a similar rate. However, evaporation is more dependent on light
quantity which will decrease with the extinction of aerosols. As a result, the weakened
evaporation in part offsets the increased transpiration, leading to the smaller growth rate
of ET than the responses of photosynthesis and the consequent enhancement in water
use efficiency (Wang et al., 2023). The iMAPLE model reasonably captures the lower
growth rates of ET than GPP in response to diffuse radiation at the selected sites.


**4. Conclusions and discussion**
We develop the iMAPLE model by coupling Noah-MP water module with YIBs
vegetation model. Validations show that iMAPLE predicts reasonable distribution of
soil moisture and soil temperature. Driven with these prognostic soil conditions and
meteorology from reanalyses, the model reasonably reproduces the observed
spatiotemporal variations of both GPP and ET fluxes at 201 sites and on the global scale.





We further update the biogeochemical processes in iMAPLE to extend the model's capability in quantifying interactions between air pollution and land ecosystems. The model reasonably predicts wetland $CH_4$ emissions at 44 sites and yields the similar global map of $CH_4$ emissions compared to an ensemble of 13 biogeochemical models. In addition, predicted biomass burning and biogenic emissions are consistent with either satellite retrievals or results from other models. We assess the impacts of surface $O_3$ and aerosols on ecosystem fluxes. The LMA-based scheme links the $O_3$ sensitivity with vegetation LMA and predicts a global map of GPP loss that is consistent with the traditional scheme using the PFT-specific sensitivity. The updated scheme effectively reduces modeling uncertainties by decreasing the number of parameters for $O_3$ sensitivity and provides an option to apply the advanced LMA map from remote sensing. The model also reproduces the observed responses of GPP and ET to diffuse radiation with a lower growth rate for ET than GPP.

There are several limitations in the current version of iMAPLE model. First, it does not include the dynamic nutrient cycle. Although we implement the down regulation from Arora et al. (2009) to constrain $CO_2$ fertilization, this limitation is dependent only on the ambient $CO_2$ concentrations and could not represent the heterogeneous distribution of nutrients. As a result, the model could not reveal the biogeochemical effects of nitrogen and phosphorus deposition on land ecosystems. Second, the feedback of fire activities to ecosystems is ignored. The iMAPLE considers the impacts of fuel load on area burned at each modeling time step. However, these fire perturbations do not in turn change the vegetation distribution and composition. The vegetation model does not consider the competition among PFTs, so that fire perturbations are not allowed to change vegetation coverage. As a result, the interactions between fire and ecosystems are underestimated in the current model framework. Third, iMAPLE does not consider the dynamic changes in wetland area for $CH_4$ emissions. Although the Noah-MP module predicts runoff and underground water, the changes of hydrological cycles are not connected with wetland aera in the model. Instead, a prescribed wetland dataset is





applied to reduce the possible uncertainties but meanwhile refrain the explorations of
$CH_4$ changes in the historical and future periods. These limitations will be the focuses
of model development in the next step.

The iMAPLE model inherits the good capability of the original YIBs model in the
simulations of carbon cycle. Furthermore, the iMAPLE upgrades the YIBs model with
carbon-water coupling and more biogeochemical processes. With the iMAPLE model,
we could assess the changes of carbon and water fluxes, as well as their coupling, in
response to environmental perturbations (e.g., climate change, air pollution, land cover
change). Meanwhile, by coupling the iMAPLE with climate and/or chemical models,
we could further quantify the changes of meteorology and atmospheric components in
response to the biogeochemical and biogeophysical processes. For example, Lei et al.
(2022) revealed the strong vegetation feedback to global surface $O_3$ during the drought
periods using the YIBs model coupled to a chemical transport model. Xie et al. (2019)
found a significant increase in atmospheric $CO_2$ concentrations due to $O_3$-induced
vegetation damage using the YIBs model coupled with a regional climate-chemistry
model. Gong et al. (2021) estimated a surface warming in polluted regions due to the
ozone-vegetation feedback using the YIBs model coupled with a global climate-
chemistry model. These studies indicate that the iMAPLE model could be used either
offline or online with other models to explore the interactions among climate, chemistry,
and ecosystems.

*Acknowledgment*. This work was jointly supported by the National Key Research and
Development Program of China (grant no. 2019YFA0606802), the National Natural
Science Foundation of China (grant no. 42275128).

*Author contributions*. XY, HL designed the research and wrote the paper. XY, HaZ
optimized codes, performed simulations, and analyzed results. XY, HaZ, CT, YM, YH,
CG implemented codes and collected data. HuZ helped with code implementations. All





authors commented on and revised the manuscript.

*Competing interests.* The contact author has declared that none of the authors has any
competing interests.

*Code availability.* The code for the iMAPLE version 1 model is available at
https://doi.org/10.6084/m9.figshare.23593578.v1

*Data availability.* All the validation data are available to download from the cited
references or data links shown in Section 2.4. The simulation data of monthly output
from BASE experiment during 1980-2021 with the iMAPLE model are available at
https://doi.org/10.6084/m9.figshare.23593578.v1

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

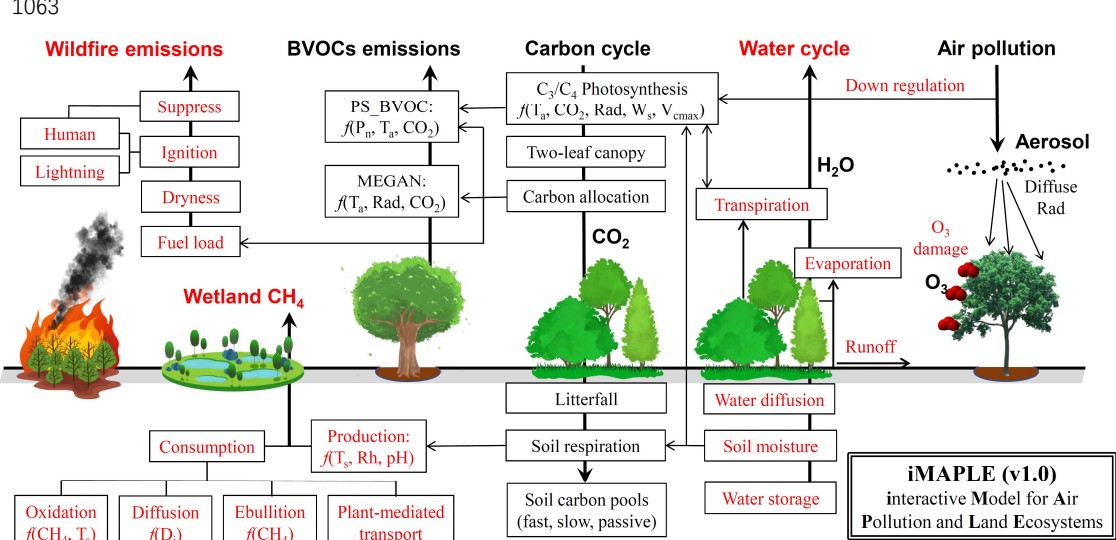


**Figure 1** The illustration of biogeochemical processes in the iMAPLE version 1.0
model. The carbon cycle is connected with water cycle, wildfire emissions, biogenic
volatile organic compounds (BVOCs) emissions, wetland methane emissions, and is
affected by air pollutants including aerosols and ozone. The bold arrows indicate the
directions of fluxes and air pollutants. The thin arrows indicate the influential pathways
among different components. The dependences on key parameters are shown for some
processes. Red fonts indicate new or updated processes in iMAPLE relative to the YIBs
model. For detailed parameterizations please refer to section 2.2.

1074
1075
1076



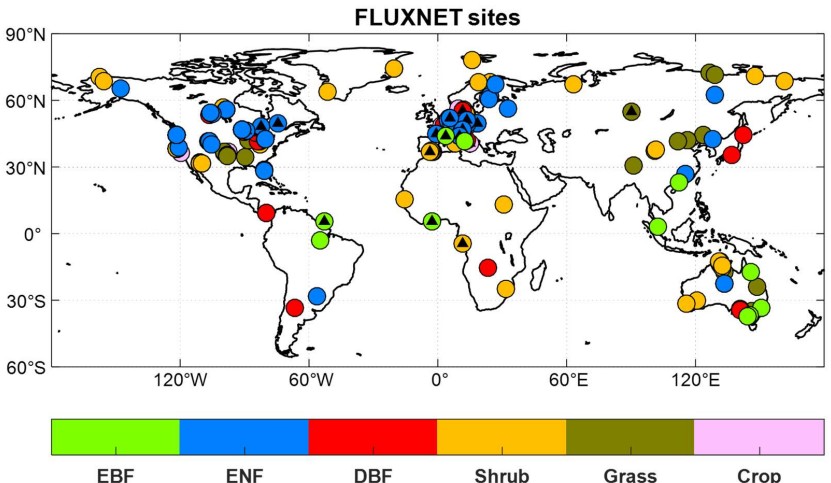

**Figure 2** Spatial distributions of 201 sites from global FLUXNET network. The colors indicate various plant functional types (PFTs) including evergreen broadleaf forest (EBF, 13 sites), evergreen needleleaf forest (ENF, 57 sites), deciduous broadleaf forest (DBF, 25 sites), Shrub (52 sites), Grass (37 sites), and Crop (17 sites). The black triangles indicate sites with at least one-year observations of diffuse radiation.

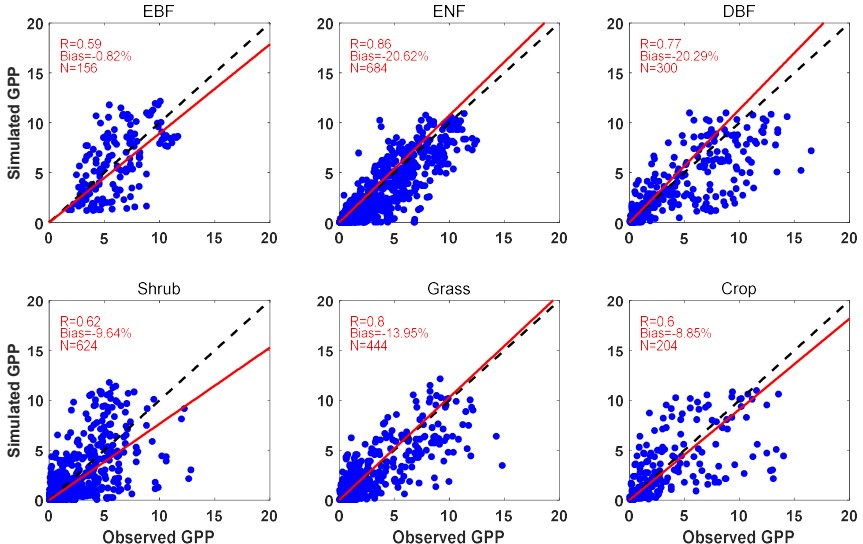

**Figure 3** Comparisons between observed and simulated monthly GPP from 201 FLUXNET sites. Each point indicates the average value of one month at a site. The red line represents linear regression between observations and simulations. The correlation coefficient (R), normalized mean bias and numbers of points/months (N) are shown on each panel. The comparisons are grouped into six PFTs including EBF, ENF, DBF, Shrub, Grass, and Crop. The unit is g C m$^{-2}$ day$^{-1}$.





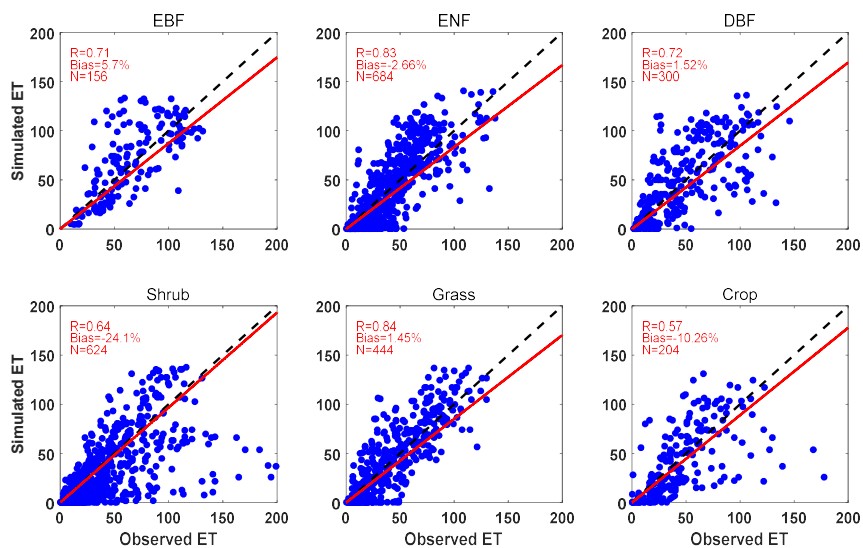


**Figure 4** The same as Figure 3 but for ET. The unit is mm month$^{-1}$.


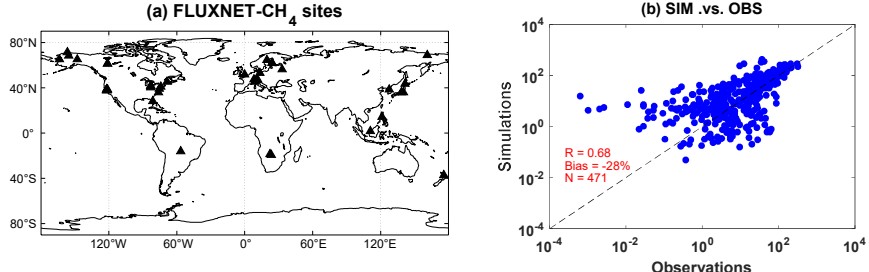

**Figure 5** Spatial distribution of global FLUXNET-CH₄ sites and comparisons between observed and simulated monthly methane flux. Filled triangles indicate sites with at least six months observations of wetland CH₄ fluxes. Each point represents average value of monthly methane emission at one site. The correlation coefficient (R), normalized mean bias and numbers of points/months (N) are shown on the right panel. The unit is g [CH₄] m$^{-2}$ yr$^{-1}$.



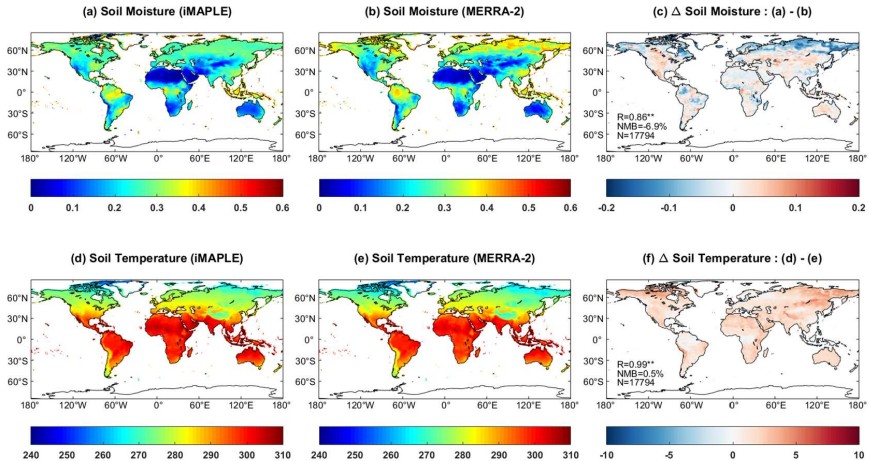


**Figure 6** Comparisons of simulated (a) soil moisture ($m^3$ $m^{-3}$) and (d) soil temperature (K) from the
iMAPLE model with (b, e) the MERRA-2 reanalyses. Both simulations and observations are
averaged for the period of 1980-2020. The spatial difference, correlation coefficient (R), normalized
mean bias (NMB) between simulations and observations and numbers of points (N) are shown on
(c) and (f), respectively.



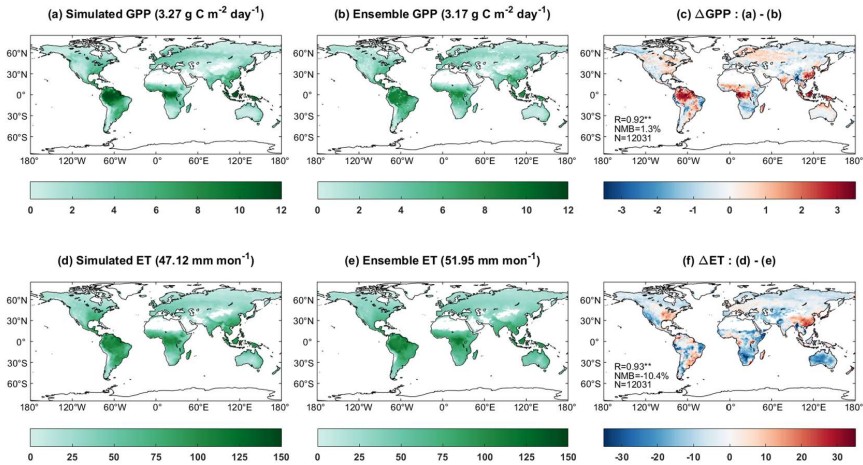

**Figure 7** Comparisons of simulated (a) gross primary productivity (GPP, g C m$^{-2}$ day$^{-1}$) and (d) evapotranspiration (ET, mm month$^{-1}$) with ensemble products from (b, e) observations. Simulated GPP and ET are performed by iMAPLE driven with meteorology from MERRA-2 reanalysis during 2001-2013. Ensemble GPP products are from the average values of SIF-based GOSIF and satellite-based GLASS GPP products. Ensemble ET products include FLUXCOM and GLASS products during 2001-2013. The spatial difference, correlation coefficient (R), normalized mean bias (NMB) between simulations and observations and numbers of points (N) are shown on (c) and (f). Only land grids with vegetation are shown on each panel, and their area-weighed values are shown in titles.





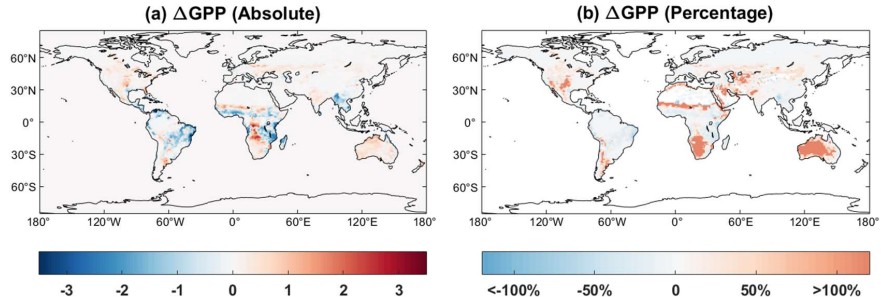

**Figure 8** Absolute (g C m$^{-2}$ day$^{-1}$) and relative (%) differences of global GPP between simulations with and without two-way carbon-water coupling processes. Simulation results are averaged for the period of 1980-2020.



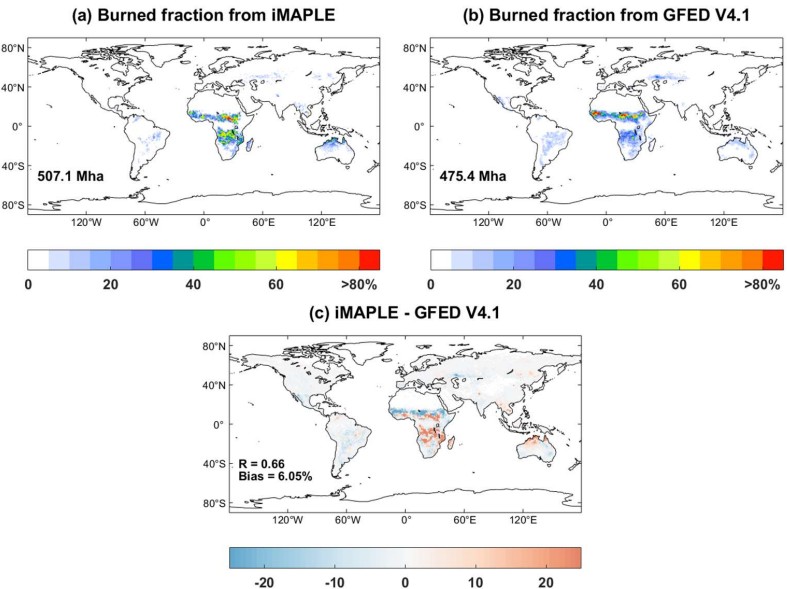


**Figure 9** Comparisons of global burned fraction (%) between (a) simulations and (b) observations. Simulations are performed using iMAPLE and observations are from GFED V4.1 fire emissions products. Both simulations and observations are averaged for the 1997-2016 period. The global total area burned are shown on (a) and (b). The spatial difference, correlation coefficient (R), and normalized mean biases between simulations and observations are shown on (c).


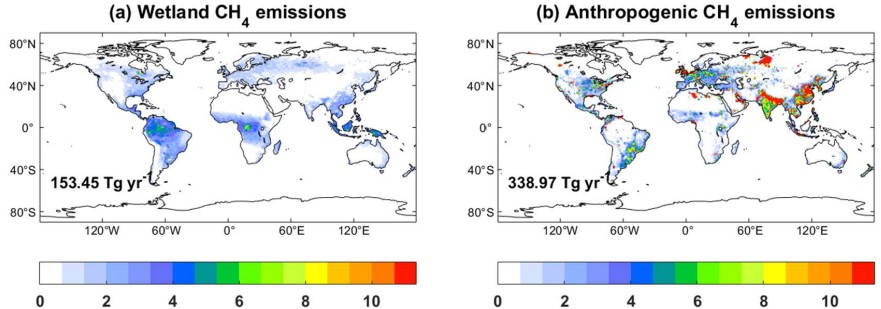

**Figure 10** Global CH$_4$ emissions (g [CH4] m$^{-2}$ yr$^{-1}$) from (a) wetland and (b) anthropogenic sources. Anthropogenic sources include energy, agriculture, industrial, residential, shipping, solvent and transportation. The global total emissions are shown on each panel. Both the wetland and other emissions are averaged for 2000-2014.

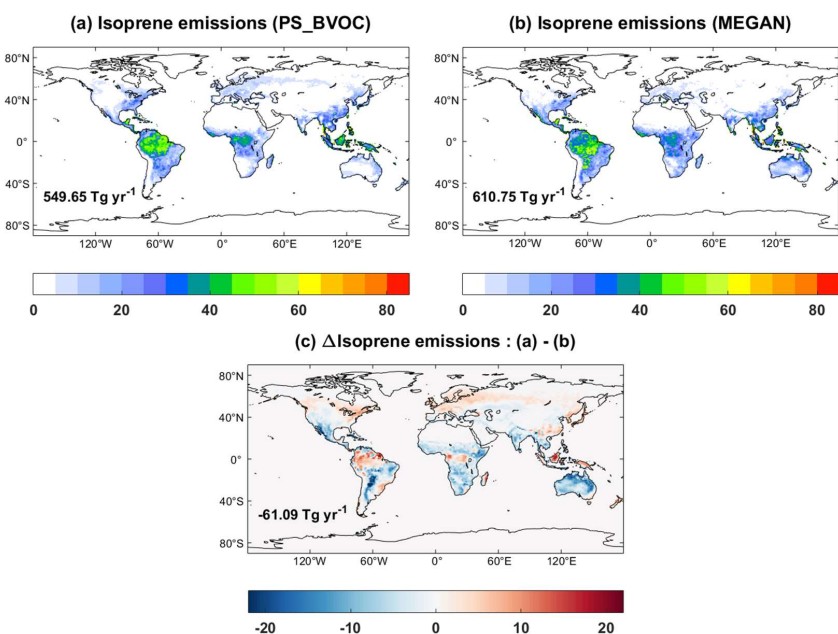

**Figure 11** Global isoprene emissions (mg C m$^{-2}$ day$^{-1}$) from (a) MEGAN, (b) PS_BVOC schemes
and (c) their differences. The global total emissions are shown on each panel.




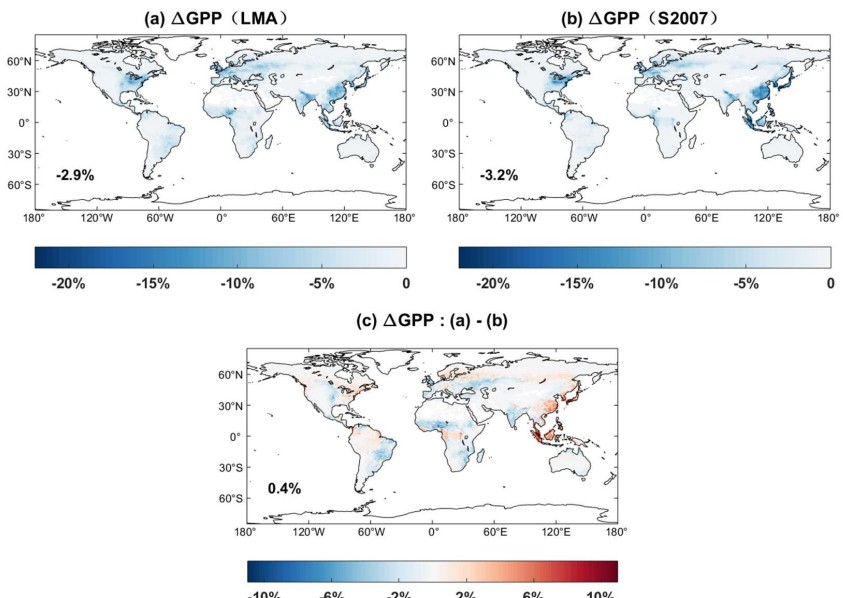


**Figure 12** Percentage changes of global GPP caused by ozone damage effects. The ozone damage


schemes include (a) trait leaf mass per area (LMA)-based, (b) S2007 plant ozone sensitivity and (c)


their differences.





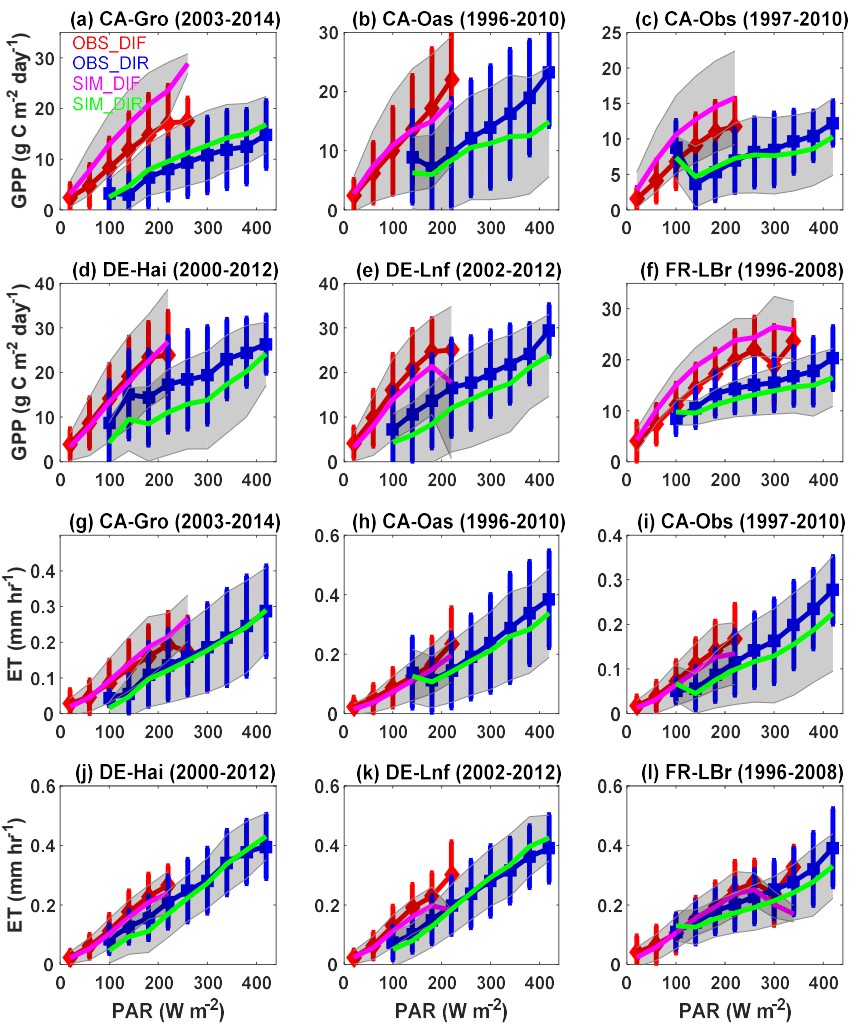

**Figure 13** Observed and simulated responses of site-level (a-f) GPP and (g-l) ET to diffuse and direct radiation at the FLUXNET sites. Photosynthetically active radiation (PAR) reaching the surface are divided into diffuse (diffuse fraction > 0.75) and direct (diffuse fraction < 0.25) radiation at six FLUXNET sites with more than 10 years of observations. Observations (simulations) are grouped over PAR bins of 40 W m$^{-2}$ with errorbars (shadings) indicating standard deviations of GPP and ET for each bin. The red (blue) and magenta (green) represent observed and simulated responses of GPP and ET to diffuse (direct) radiation. Units of GPP and ET are g C m$^{-2}$ day$^{-1}$ and mm hr$^{-1}$, respectively.