# Peer review of "Development and evaluation of the interactive Model for Air Pollution and Land"

_Geoscientific Model Development, 2023_

## Author Comment (AC1)

*Review1#:*

The authors are grateful to the editor and two reviewers for their time and energy in providing helpful comments that have improved the manuscript. In our revised paper, we added more explanations on model parameterizations (e.g., $CH_4$ emissions) and improved descriptions on Figures (e.g., Figure 9) and Tables to help readers understand our manuscript better.

In this document, reviewers' comments have been addressed point by point. Referee comments are shown in black italics and author responses are shown in blue regular text. A manuscript with tracking changes is submitted separately.

*General comments:*

*This manuscript describes the development and validation of the interactive Model for Air Pollution and Land Ecosystems (iMAPLE). This involves coupling the process-based water cycle module from Noah-MP to an updated version of the Yale Interactive terrestrial Biosphere (YIBs) model.*

*The manuscript is well written, provides a comprehensive documentation of the development work, and includes a substantial expansion of the observations used in the evaluation of earlier versions of the YIBs model.*

➔We thank the reviewer for the positive evaluations.

*Specific comments:*

*In Section 2.3, the simulations performed are described and called "BASE", "BASE_NW", "O3LMA" and "O3S2007" but they are not consistently referred to using these names during the rest of the manuscript. It would aid the reader if the simulation names were used to refer to them throughout, and in Figure captions.*

➔Thanks for your suggestions. We added more references of simulations name on the manuscript (e.g., Lines 564-565) and Figure captions (e.g., Figures 3-12).

*Line 56: could you be more specific here than "the ecosystem"*

➔We corrected original descriptions using "the terrestrial ecosystem".

*Line 57 – 61: as the size of the estimated net carbon sink is not constant over time (which you mention later in the Introduction) can you state a time period for the 2 Pg C yr$^{-1}$ value quoted here?*

➔We added specifical time periods on our descriptions as follows:
"leading to a net carbon sink of only ~2 Pg C yr$^{-1}$ during 1960-2021 (Friedlingstein et al., 2022)." (Lines 64-65)

*Line 212: Ws is mentioned here but I don't think it's defined (apologies if I missed that) and it's not currently clear how this relates to equation 7, could you clarify – perhaps it should be Wsoil?*

➔Yes, we corrected $W_S$ to $W_{soil}$ in the revised manuscript.

*Line 261: this is slightly confusing because "U" is defined in the sentence previously but "UP" is included in equation 20 and not yet defined. Could you rearrange the text to clarify?*

➔We rearranged the descriptions to clarify these equations as follows:
"The burned area of a single fire ($BA_{single}$) is typically taken to be elliptical in shape associated with length-to-breadth ratio ($LB$), head-to-back ratio (HB) and rate of fire spread ($UP$) as follows:

$$BA_{single} = \frac{\pi \times UP^2}{4 \times LB} \times (1 + \frac{1}{HB})^2 \qquad (20)$$

Then, LB and HB are related to changes of near-surface wind speed ($U$) as follows:

$$LB = 1 + 10 \times (1 - e^{-0.06 \times U}) \qquad (21)$$

$$HB = \frac{LB + (LB^2 - 1)^{0.5}}{LB - (LB^2 - 1)^{0.5}} \qquad (22)$$

Meanwhile, $UP$ is computed as the function of relative humidity ($RH$):

$$UP = UP_{max} \times f_{RH} \times f_\theta \times G(W) \qquad (23)$$

Here, $UP_{max}$ is the maximum fire spread rate depending on PFTs" (Lines 284-293)

*Line 266: I dont think f$_{RH}$ and f$_\theta$ are defined*

➔In our revised manuscript, we defined f$_{RH}$ and f$_\theta$ as follows:
"f$_{RH}$ and f$_\theta$ represent the dependence of fire spread on RH and on root-zone soil moisture, respectively." (Lines 293-294)

*Line 328 – 330: could you expand on this, which plant related factors determine A$_{CH4}$ in the model, is it parameterised?*

➔In the revised paper, we added more explanations on parameters of $A_{CH4}$ as follows:

"The plant-mediated transport of $CH_4$ through aerenchyma is dependent on the concentration gradient of $CH_4$ and the plant-related factors (Zhu et al., 2014). The $A_{CH4}$ is determined by the oxidation factor of root and the aerenchyma factor of plant. The baseline value of the oxidation factor in root is 0.5, with a regulatory range from 0.2 to 1.0 determined by the types of plant in wetland. The plant aerenchyma factor is calculated by the ratio of plant root length density (typical value: 2.1 cm mg-1) and root cross-sectional area (typical value: 0.0013 cm2), along with the diffusion factor of methane from plant root to atmosphere which is modulated by plant species within a range of 0 to 1 (Zhang et al., 2002)." (Lines 379-387)

*Line 399: where do the surface $O_3$ concentrations required for the parameterisations come from (in the absence of coupling to an atmospheric chemistry model)?*

➜We added more descriptions on surface $O_3$ concentrations as follows:
"Surface hourly $O_3$ concentrations are adopted from the simulations with a chemical transport model used in our previous study (Yue and Unger, 2018)." (Lines 439-441)

*Lines 424 – 431: would this description of the observations be better placed in Section 2.4 above?*

➜Thanks for your suggestions. We moved this part into Section 2.4.

*Line 436: 438: are you basing the point that iMAPLE improves GPP simulations as compared to YIBs on simulations presented here (i.e. BASE_NW) or referring to previously published evaluations of YIBs? If the former can you refer to any figures that demonstrate this, if the latter can you include any comparable statistics from previous work?*

➜Thanks for your questions. We found that iMAPLE with coupled water cycle improved GPP simulations compared to previous evaluations using YIBs model in YU2015, and we further clarified this information and provided comparable statistics in the revised paper as follows:
"Compared to previous evaluations from the YIBs model (YU2015), iMAPLE with coupled water cycle improves the R of GPP simulations for ENF (from 0.65 to 0.86) and grassland (from 0.7 to 0.8) but worsens the predictions for other species such as EBF (from 0.65 to 0.59)." (Lines 494-498)

*Line 462: should the second site mentioned here be US-Tw4 (as referenced in the next sentence)? Could you also include here what the simulated $CH_4$ flux is for the gridcell that contains these two sites, for the corresponding time period? It would be useful for the reader to understand whether the simulated value lies somewhere between the two observed values or not.*

➔Thanks for your questions. We corrected US-Tw4 in the revised paper, and added more descriptions on simulated $CH_4$ flux as follows:

"For example, US-Tw1 and US-Tw4 are two nearby sites within a distance of 1 km, where our simulations present $CH_4$ flux of 14.35 $g[CH_4]$ $m^{-2}$ $yr^{-1}$ during 2011-2017. However, average $CH_4$ flux shows a difference of 3.7 times with 66.31 $g[CH_4]$ $m^{-2}$ $yr^{-1}$ in US-Tw1 and 18.16 $g[CH_4]$ $m^{-2}$ $yr^{-1}$ in US-Tw4 during 2011-2017." (Lines 523-527)

*Lines 554 - 565: This section describes the impact of $O_3$ damage on GPP under 2 different schemes but it would benefit from some clarity around the level of $O_3$ damage being simulated. I think panel (a) must represent the difference between GPP in the O3LMA simulation and the BASE simulation, but this needs to be stated in the discussion and Figure 12 caption. This is important because you go on to compare the impact on GPP to the value from Ma et al 2023 but it is not currently clear if the two % values are really comparable.*

➔ Following this suggestion, we added more descriptions of simulations caused by two experiments on Figure 12 caption.

*Lines 568 - 571: is this based on separating the FLUXNET or MERRA-2 shortwave radiation into diffuse and direct? It would be useful to add a note here to clarify that.*

➔As suggested by the reviewer, we added descriptions on observed FLUXNET diffuse and direct radiation as follows:

"Here, we separate the diffuse (diffuse fraction > 0.75) and direct (diffuse fraction < 0.25) components using observed diffuse fraction and solar radiation at six FLUXNET sites, and aggregate the GPP and ET fluxes for different radiation periods at certain intervals (Figure 13)." (Lines 643-646)

*Line 1086: specify in the caption for Figure 3 that this data is from the BASE simulation (if it is) - this suggestion applies to all Figures*

➔Thanks for your suggestions. We added the names of simulations to all Figures.

*Line 1097: refer to panels (a) and (b) in the caption. Can you label the axes in panel (b) to specify that these are observed / simulated $CH_4$ fluxes, with units.*

➔Corrected as suggested.

*Line 1104: please label the colour bars in Figure 6, or add the units to the title of each panel*

➔Added as suggested, and we added the units into the title of each panel.

*Line 1136: add to this caption that the anthropogenic emissions are taken from CMIP6 input (rather than being generated by iMAPLE)*

➔Added as suggested.

*Line 1142: specify the time period that the emissions represent. Assuming these are annual totals, do they represent the entire simulation period?*

➔Added as suggested.

***Technical corrections:***
*Line 58: "these" should be "this" and "respirations" should be "respiration"*

➔Corrected as suggested.

*Line 100: "matters" should be "matter"*

➔Corrected as suggested.

*Line 105: "assimilations" should be "assimilation"*

➔Corrected as suggested.

*Line 122: "BVOCs" should be "BVOC"*

➔Corrected as suggested.

*Line 393: "lighting" should be "lightning"*

➔Corrected as suggested.

*Line 429: "much" should be "many"*

➔Corrected as suggested.

*Line 544: I think "we" should be "as"*

➔Corrected as suggested.

---

## Author Comment (AC2)

*Review2#:*

The authors are grateful to the editor and two reviewers for their time and energy in providing helpful comments that have improved the manuscript. In our revised paper, we added more explanations on model parameterizations (e.g., $CH_4$ emissions) and improved descriptions on Figures (e.g., Figure 9) and Tables to help readers understand our manuscript better.

In this document, reviewers' comments have been addressed point by point. Referee comments are shown in black italics and author responses are shown in blue regular text. A manuscript with tracking changes is submitted separately.

*Review of Development and evaluation of the interactive Model for Air Pollution and Land Ecosystems (iMaple) version 1.0 by Xu Yue et al.*
*The paper describes a substantial upgrade to the YIBs model, the paper is well laid out and generally has an appropriate amount of detail on the model description and evaluation. I agree with the final conclusion that the new model is well suited for studying climate-chemistry-ecosystem interactions, either driven by atmospheric data or coupled to an atmospheric model. There are several places where the text could be clearer or a little more detail could be added, I list these below.*

➔Thanks for your positive comments, and we added more details as suggested in the revised paper.

*General: Please can you add a table that lists all parameters, their values and units?*

➔In our revised paper, we listed all parameters into Tabel S1 as suggested.

*Line 143: In section 2.1, please can you state if each gridbox has a single PFT, or a mixture of PFTs? If it is a mixture of PFTs, do all PFTs share a single soil column (all draw from the same soil moisture store)?*

➔In the revised paper, we added more explanations on PFTs in the model as follows: "At each grid, a mixture of PFTs with each PFT fraction is used as model input, sharing the temperature or moisture information from the same soil column." (Lines 155-157)

*Line 184: How are ECAN and EGRO calculated? TWS doesn't include a canopy storage term, so what storage term is the ECAN flux taken from?*

➔In our revised paper, we added the equations on calculations of ECAN and EGRO, and explained that ECAN fluxes are from canopy interception of precipitation as follows:

"Meanwhile, $ECAN$ is calculated as follows:

$$ECAN = \frac{\rho_{air} \cdot CP_{air} \cdot C_{canopy,evap} \cdot (e_{sat} - e_{ca})}{PC} \tag{4}$$

$$C_{canopy,evap} = \frac{f_{wet} \cdot E_{VAI}}{R_{leaf,bdy}} \tag{5}$$

Here, $C_{canopy,evap}$ is the latent heat conductance from the wet leaf surface to canopy air. $f_{wet}$ is the wetted fraction of canopy, which is a fraction of the maximum canopy precipitation interception capacity. $E_{VAI}$ is the effective vegetation area index and $R_{leaf,bdy}$ is bulk leaf boundary resistance. $EGRO$ is calculated as follows:

$$EGRO = C_{ground,evap}(e_{sat,ground}RH - e_{ca}) \tag{6}$$

Here, $C_{ground,evap}$ is the coefficient for latent heat, $e_{sat,ground}$ is the saturated vapor pressure at the ground and RH is the surface relative humidity." (Lines 202-211)

*Line 200: "Here, we assume independent and exponential distributions of infiltration capacity and precipitation in each grid cell when considering soil infiltration processes and Qsoil,in is the infiltration into the soil, following the approach by Schaake et al. (1996)." I don't understand this sentence, please can you expand on what you mean. Do you mean each grid cell is independent of all other grid cells? Or that infiltration capacity is independent of precipitation? Do you mean that there is an exponential relationship between infiltration capacity and precipitation?*

➜In our revised paper, we expanded original descriptions and further added equations on calculations of infiltration capacity in each grid cell as follows:
"$Q_{soil,in}$ is the infiltration into the soil, which is derived from approximate solutions of Richards equations with considerations of the spatial variations in precipitation and infiltration capacity. Here, we assume exponential distributions of infiltration capacity in each grid cell following the approach by Schaake et al. (1996):

$$Q_{soil,in} = Q_{soil,srf} \frac{I_c}{Q_{soil,srf}\Delta t + I_c} \tag{9}$$

$$I_c = W_d[1 - \exp(-K_{\Delta t}\Delta t)] \tag{10}$$

Here, $I_c$ and $W_d$ are the soil infiltration capacity of the model grid cell and the water deficit of the soil column, respectively. $K_{\Delta t}$ and $\Delta t$ are the calibratable parameters and model time step." (Lines 218-227)

*Line 206: Does K4 vary spatially? If it does vary spatially, what dataset is used to calculate K4?*

➜In our revised paper, we added more explanations on calculating $K_4$ as follows:

"K$_4$ is the hydraulic conductivity in the bottom soil layer parameterized following the scheme in Clapp and Hornberger (1978) and is calculated using spatial soil profiles from Hengl et al. (2017)." (Lines 230-232)

*Line 215: The model's soil column is only 2m deep, was that depth inherited from Noah-MP, or chosen by the authors? I think a deeper soil column would improve the ecosystem representation and interactions with climate, particularly during drier conditions. Perhaps, you could comment on the choice of soil of total soil depth in the discussion.*

➔Thanks for your questions. The soil column in iMAPLE model is inherited from original Noah-MP model, and we added more discussion in the last section as follows: "Meanwhile, iMAPLE model considers only dynamic soil water and temperature at 2-m level, which may influence the deeper soil interactions between climate and land terrestrial ecosystem especially for the drier conditions." (Lines 709-711)

*Line 240: In section 2.2.2, please state that simulated burnt area has not impact on vegetation, or feedback onto fuel load. I appreciate this is mentioned in the discussion, but I think it should be mentioned here too.*

➔In our revised paper, we added more descriptions on fire modules of iMAPLE model as follows:
"It is important to note that the feedbacks of fire activities on terrestrial ecosystems have not been considered in the current version of iMAPLE model due to the high complexity." (Lines 313-315)

*Line 254: What are the units of PD?*

➔The unit of PD is Number km$^{-2}$, and units of all parameters used in this study are shown in Table S1.

*Line 261: This function is complicated, it is difficult to know how the burnt area is related to the atmospheric drivers. Please can you plot BA as a function of U for a fixed RH, and plot BA as a function of RH for a fixed U? If plots of these relationships exist in Pechony and Shindell (2009) or Li et al. (2012), you could refer to their figures, but it would still be good to add some description of the relationships, e.g. does BA depend strongly on U, is RH more important, what happens if U is zero?*

➔In our revised paper, we plotted relationships between burned area (BA) and U, RH as Figure R1, and added more explanations on relationships between burned area and atmospheric drivers (e.g., wind speed and relative humidity) as follows:

[Figure]

[Figure]

**Figure R1.** The dependences of BA_single on (a) near-surface wind speed (U) and (b) relative humidity (RH), respectively.

"In general, the eccentricity of burned area is primarily influenced by near-surface wind speed, while the rate of fire spread is jointly regulated by near-surface wind speed and relative humidity. The shape of the fire is converted to a circular form when the near-surface wind speed reaches zero, and burning ceases to propagate once the relative humidity is above a specific threshold." (Lines 302-306)

*Line 270: Why were these values of RHlow and RHup chosen?*

➔In our revised paper, we explained reasons for choosing $RH_{low}$ and $RH_{up}$ as follows: "In this study, we set $RH_{low}$ =30 % and $RH_{up}$ =70 % as the lower and upper thresholds of RH following the methods used in Li et al. (2012). If RH is higher than 70%, natural fires will not occur or spread, and RH will no longer be a constraint factor for fire occurrence and spread if RH ≤ 30%." (Lines 297-300)

*Line 298: Should the left hand side of equation 30 be "fTS" and not "Q10"?*

➔Thanks for your questions, and we added formula on calculating $F_{TS}$ to avoid misunderstanding as follows:

"The impact factor of soil temperature $f_{ST}$ can be calculated as follows (Zhang et al., 2002; Zhu et al., 2014):

$$f_{ST} = \begin{cases} 0, & T_s < T_{min} \\ vt^{xt} \exp(xt(1-vt)), & T_{min} \leq T_s \leq T_{max} \\ 0, & T_s > T_{max} \end{cases} \quad (35)$$

$$vt = (T_{max} - T_s)/(T_{max} - T_{opt}) \quad (36)$$

$$xt = \left[\log(Q_{10})\left(T_{max} - T_{opt}\right)\right]^2 (1.0 + at^{0.5})^2/400.0 \quad (37)$$

$$at = 1.0 + 40.0/[\log(Q_{10})(T_{max} - T_{opt})] \quad (38)$$

$T_{min}$, $T_{max}$, and $T_{opt}$ represents the lowest, highest and optimum temperature for the process of methane production and oxidation, respectively. In this study, the $T_{min} = 0°C$, $T_{max} = 45°C$ and $T_{opt} = 25°C$ (Zhu et al., 2014)." (Lines 336-345)

*Line 363: How do you solve equations 36 and 38? Do you use an iterative process?*

➔Thanks for your questions. The original equations 36 and 38 are now 45 and 47 in the revised paper, respectively. We clarified as follows:
"The $f_{O3}$ in Equation (45) is fed into Equation (47) so as to build a quadratic equation for F. We solve the quadratic equation and select the *F* value within the range of [0, 1]." (Lines 420-422)

*Line 385: How is the spin-up run? What driving data is used?*

➔In our revised paper, we clarified the spin-up processes as follows:
"We run the model for the period of 1980-2021 using the initial conditions of the equilibrium soil carbon pool, tree height, and water fluxes from a spin-up run of 200 years driven with cycled forcing at the year 1980." (Lines 444-447)

*Line 399: Is LMA PFT-specific? Are you using a different map of LMA for each PFT?*

➔In our model, the LMA map has no PFT information but shows specific values for each individual grid. We clarified as follows:
"For the LMA-based $O_3$ damage scheme, we use gridded LMA from the trait-level dataset of TRY (Kattge et al., 2011) developed by extending field measurements with the random forest model (Moreno-Martínez et al., 2018)." (Lines 461-464)

*Line 418: Please state that CMIP6 anthropogenic CH4 emissions are used for context and not for validation purposes.*

➔In our revised paper, we moved this part into the Section 2.2.3 as follows:
"We implement the process-based wetland $CH_4$ emissions into the iMAPLE model. The anthropogenic sources of $CH_4$ from Coupled Model Intercomparison Project phase 6 (CMIP6, https://esgf-node.llnl.gov/projects/input4mips/) are also used as input for iMAPLE." (Lines 318-321)

*Line 440: In addition to biases in meteorological input, it would be good to acknowledge that 1x1 degree gridded simulations would not be expected to match site-level observations, because of differences in vegetation cover and soil properties.*

➔In our revised paper, we added further discussion on biases of simulations as follows: "The biases in the meteorological input may cause uncertainties in the simulation of GPP fluxes (Ma et al., 2021). In addition, the mismatch of vegetation cover and soil properties between the site location and 1°×1° grid in the simulation may further contribute to the modeling biases." (Lines 500-503)

*Line 442: Why would the increase in site number and record length decrease R?*

➔Considering these inappropriate descriptions, we deleted this sentence in our revised paper.

*Line 482: In this paragraph, it would be good to include the global total GPP in Pg/yr. This would allow easy comparison to other estimates.*

➔We added global total GPP using the units of Pg $yr^{-1}$ in our revised paper as follows: "On the global scale, our simulations yield a total GPP of 129.8 Pg C $yr^{-1}$, similar to the observed amount of 125.4 Pg C $yr^{-1}$." (Lines 549-550)

*Line 512: This paragraph and figure 9 would be better if fire emissions were evaluated. The paper and the model have a focus on climate-chemistry-ecosystem interactions, and fire has been included in order to simulate emissions, not particularly to predict burnt area.*

➔ Thank you for your suggestions. In the revised paper, we added evaluations of simulated fire-emitted OC emissions using GFED products in Figure 9 and more descriptions as follows:
"Furthermore, we compare fire-emitted OC from the model with GFED4.1s. The spatial pattern of OC emissions is similar to that of burned area. The simulations yield a total of 16.8 Tg $yr^{-1}$ for the global fire-emitted OC, slightly higher than the amount of 16.4 Tg $yr^{-1}$ from GFED4.1s with some overestimations in tropical Africa (Figure 9f)." (Lines 591-595)

[Figure]

**Figure 9** Comparisons of global burned fraction (%) and fire-emitted OC emissions (10⁻³ kg km⁻¹ yr⁻¹) between (a, d) simulations and (b, e) observations. Simulations are performed using iMAPLE and observations are from GFED V4.1 fire emissions products. Both simulations from BASE experiment and observations are averaged for the 1997-2016 period. The global total area burned are shown on (a) and (b), and total OC emissions are shown on (d) and (e). The spatial difference, correlation coefficient (R), and normalized mean biases between simulations and observations are shown on (c) and (f).

*Line 624: "As a result, the interactions between fire and ecosystems are underestimated in the current model framework." Can you speculate on how these interactions would affect your results? My guess is that the lack of feedback from burnt area to fuel load, means that the model will overestimate burnt area and fire emissions.*

➔In our revised paper, we added discussions on fire feedbacks as follows:
"As a result, the interactions between fire and ecosystems are underestimated in the current model framework, potentially leading to overestimations of wildfire activity due to remaining fuel loads." (Lines 701-704)

*Line 629: Instead of "refrain", perhaps "limits" or "prevents" would be better words to use.*

➔Corrected as suggested.

*Figure 6: Could consider using a colourblind friendly colour scale. I don't know if the journal has a policy on this.*

➔Corrected as suggested.

*Figure 9: Would be improved by including a maps of fire emissions from iMAPLE and GFED. I'm not sure what emissions data GFED provides, but emissions of a single species would be sufficient.*

➔Thanks for your suggestions. We added validations of simulated fire-emitted OC with GFED products into Figure 9 and more descriptions as follows:
"Furthermore, we compare fire-emitted OC from the model with GFED4.1s. The spatial pattern of OC emissions is similar to that of burned area. The simulations yield a total of 16.8 Tg yr$^{-1}$ for the global fire-emitted OC, slightly higher than the amount of 16.4 Tg yr$^{-1}$ from GFED4.1s with some overestimations in tropical Africa (Figure 9f)." (Lines 591-595)

[Figure]

**Figure 9** Comparisons of global burned fraction (%) and fire-emitted OC emissions ($10^{-3}$ kg km$^{-1}$ yr$^{-1}$) between (a, d) simulations and (b, e) observations. Simulations are performed using iMAPLE and observations are from GFED V4.1 fire emissions products. Both simulations from BASE experiment and observations are averaged for the 1997-2016 period. The global total area burned are shown on (a) and (b), and total OC emissions are shown on (d) and (e). The spatial difference, correlation coefficient (R), and normalized mean biases between simulations and observations are shown on (c) and (f).

*Figure 10: Would be improved by adding maps of soil carbon and wetland area.*

➔We understood these concerns on sources of CH$_4$ emissions. In the global natural wetlands, CH$_4$ productions are commonly associated with anaerobic conditions. Heterotrophic respiration can regulate the ratio of carbon dioxide to methane by influencing the accumulation of soil organic matter which can provide available substrate for CH$_4$ production. As a result, we added maps of wetland area and

heterotrophic respiration instead of soil carbon into Figure 10 with more descriptions as follows:

"As important factors driving CH$_4$ emissions, heterotrophic respiration shows higher values over tropical regions and eastern China with a total amount of 73.2 Pg C yr$^{-1}$ (Figure 10c), and relative high wetland coverages are found in boreal Asia and Amazon (Figure 10d)." (Lines 607-610)

[Figure]

**Figure 10** Global simulated CH$_4$ emissions (g [CH4] m$^{-2}$ yr$^{-1}$) from (a) wetland and (b) anthropogenic sources, (c) heterotrophic respiration (gC m$^{-2}$ day$^{-1}$) and (d) fraction of wetland area. Anthropogenic sources include energy, agriculture, industrial, residential, shipping, solvent and transportation from CMIP6 input. The simulations are from BASE experiment. The global total emissions and heterotrophic respirations are shown on each panel. All variables are averaged for 2000-2014.

*Figure 10: In the caption, please could you be clear that wetland emissions are simulated by iMAPLE and anthropogenic sources are taken from input4mips.*

➔In our revised paper, we added more explanations on caption in Figure 10 as suggested.

---

## Author Response (AR2)

The authors are grateful to the editor for her time and energy in providing helpful comments that have improved the manuscript. In this document, the editor's comments have been addressed point by point. All the comments are shown in black and author responses are shown in blue text. A manuscript with tracking changes is submitted separately.

Specific comments (Line numbers refer to the document with tracked changes):

1. Line 26: Change "contribute" to "contributed"

➔Corrected as suggested.

2. Experiment names: Thanks for responding to the reviewer's comment in relation to experiment names. May I please ask that you replace "from O3LMA experiment" to "from the O3MLA experiment? This change is also required for "from O3S2007 experiment", "from BASE experiment", etc. and should be applied consistently throughout the manuscript.

➔ We checked throughout the whole manuscript and made the suggested corrections.

3. Line 394: Please remove "the" in the phrase "and the plant-related factors"

➔Corrected as suggested.

4. Line 395/396: change "by the oxidation factor of root and the aerenchyma factor of plant" to "by the oxidation factor for roots and the aerenchyma factor for plants"

➔Corrected as suggested.

5. Line 396: Change "in root" to "in roots"

➔Corrected as suggested.

6. Line 397: Change "determined by the types of plant in wetland" to "determined by the wetland plant types"

➔Corrected as suggested.

7. Line 398/399: Change "ratio of plant root length density" to "ratio of the plant root length density"

➔Corrected as suggested.

8. Line 399: Change "and root cross-sectional area" to "and the root cross-sectional area"

➔Corrected as suggested.

9. Line 400: Change "the diffusion factor of methane from plant root to atmosphere" to "a plant root to atmosphere diffusion factor for methane"

➔Corrected as suggested.

10. Line 401: Change "plant species" to "plant type"

➔Corrected as suggested.

11. Line 453/454: Change "adopted from the simulations with a chemical transport model used in our previous study" to "adopted from the chemical transport model simulations used in our previous study"

➔Corrected as suggested.

12. Line 484: Change "201 sites at the FLUXNET network" to "201 sites from the FLUXNET network"

➔Corrected as suggested.

13. Line 484/485: Change "Among these sites, 95 are tree species with the major PFT of ENF" to "Among these sites, 95 have the ENF tree species as the major PFT"

➔Corrected as suggested.

14. Line 485: Change "106 are non-tree species with the maximum number for shrubland" to "106 are dominated by shrubland"

➔Since the rest 106 sites are not all shrubland, we corrected the sentence as follows: "106 are dominated by non-tree species especially shrubland".

15. Line 486: Change "of sites" to "of the sites"

➔Corrected as suggested.

16. Line 488: Change "to the earlier evaluations" to "to the previous evaluation"

➔Corrected as suggested.

17. Line 489: Change "and Southern" to "and in the Southern"

➔Corrected as suggested.

18. Line 520-521: Change "Compared to previous evaluations from the YIBs model, (YU2015)," to "Compared to the previous evaluation of the YIBs model (YU2015),"

➔Corrected as suggested.

19. Line 524: Change "cause of such deficit" to "cause of this degradation"

➔Corrected as suggested.

20. Line 551: Change "simulations present CH4 flux" to "simulations give a CH4 flux"

➔Corrected as suggested.

21. Table S1 seems to be missing!

➔The updated Table S1 is shown in the revised SI material.

22. Line 161: Replace "grid" with "gridbox"

➔Corrected as suggested.

23. Line 214: Change "wetted" to "wet"

➔Corrected as suggested.

24. Line 216: Change "is bulk leaf boundary resistance" to "is the bulk leaf boundary resistance"

➔Corrected as suggested.

25. Line 734: Change "Meanwhile, iMAPLE model" to "Meanwhile, the iMAPLE model"

➔Corrected as suggested.

26. Line 735: Change "at 2-m level" to "down to the 2-m level"

➔Corrected as suggested.

27. Line 736/737: Change "may influence the deeper soil interactions between climate and land terrestrial ecosystem especially for the drier conditions." To "may affect the interactions between climate and the land terrestrial ecosystem especially during drier conditions."

➔Corrected as suggested.

28. Line 327/328: Change "the feedbacks of fire activities on terrestrial ecosystems" to

"the feedbacks from fire activities onto the terrestrial ecosystem"

➔Corrected as suggested.

29. Line 328: When referring to iMAPLE, may I please ask that you use "the iMAPLE model" or just "iMAPLE" and not "iMAPLE model" as used here? May I also please ask that make this change consistently throughout the manuscript?

➔ We checked throughout the whole manuscript and made the suggested corrections.

30. My understanding from your response to the reviewer's comment about the complexity of original equation 20 (now equation 25) was that you were going to include a new figure. But Figure R1 seems to be missing!

➔We have put Figure R1 into SI as Figure S1. In the main text, we also added the following statement to refer to Figure S1: "The dependence of BAsingle on U and RH is shown in Figure S1."

31. Line 313: Remove the term "factor"

➔Corrected as suggested.

32. Line 460: Change "driven with cycled forcing at the year 1980" to "driven with perpetual forcing for the year 1980"

➔Corrected as suggested.

33. Line 477: Change "with the random forest model" to "with a random forest model"

➔Corrected as suggested.

34. Line 333: Change "from Coupled Model Intercomparison Project phase 6" to "from Phase 6 of the Coupled Model Intercomparison Project"

➔Corrected as suggested.

35. Line 529: Change "and 1º×1º grid" to "and the 1º×1º grid"

➔Corrected as suggested.

[revised manuscript text omitted]